# Identification of Potential Therapeutic Targets for Sensorineural Hearing Loss and Evaluation of Drug Development Potential Using Mendelian Randomization Analysis

**DOI:** 10.3390/bioengineering12020126

**Published:** 2025-01-29

**Authors:** Shun Ding, Qiling Tong, Yixuan Liu, Mengyao Qin, Shan Sun

**Affiliations:** 1ENT Institute and Otorhinolaryngology Department of Eye & ENT Hospital, State Key Laboratory of Medical Neurobiology and MOE Frontiers Center for Brain Science, Fudan University, Shanghai 200031, China; dingshunent@fudan.edu.cn (S.D.); 23211260026@m.fudan.edu.cn (Q.T.); 22111260031@m.fudan.edu.cn (Y.L.); 2Institute of Microbiology, Heilongjiang Academy of Sciences, Harbin 150000, China; qinmy802@163.com

**Keywords:** sensorineural hearing loss, UK biobank, Mendelian randomization, molecular docking

## Abstract

**Background:** Sensorineural hearing loss (SNHL) is a major contributor to hearing impairment, yet effective therapeutic options remain elusive. Mendelian randomization (MR) has proven valuable for drug repurposing and identifying new therapeutic targets. This study aims to pinpoint novel treatment targets for SNHL, exploring their pathophysiological roles and potential adverse effects. **Methods:** This research utilized the UKB-PPP database to access cis-protein quantitative trait locus (cis-pQTL) data, with SNHL data sourced from the FinnGen database as the endpoint for the MR causal analysis of drug targets. Colocalization analysis was employed to determine whether SNHL risk and protein expression share common SNPs. A phenotype-wide association analysis was conducted to assess the potential side effects of these targets. Drug prediction and molecular docking were subsequently used to evaluate the therapeutic potential of the identified targets. **Results:** Four drug target proteins significantly associated with sensorineural hearing loss (SNHL) were determined by Mendelian randomization (MR) analysis and co-localization analysis. These drug targets include LATS1, TEF, LMNB2, and OGFR and were shown to have fewer potential side effects when acting on these target proteins by phenotype-wide association analysis. Genes associated with sensorineural hearing loss are primarily implicated in the Hippo signaling pathway, cell–cell adhesion, and various binding regulatory activities and are involved in the regulation of cell proliferation and apoptosis. Next, drugs for the treatment of SNHL were screened by the DsigDB database and molecular docking, and the top 10 drugs were selected based on *p*-value. Among them, atrazine CTD 00005450 was identified as the most likely therapeutic target, followed by ampyrone HL60 DOWN and genistein CTD 00007324. In addition, LMNB2, LATS1, and OGFR could be intervened in by multiple drugs; however, fewer drugs intervened in TEF. **Conclusion:** This study has successfully identified four promising drug targets for SNHL, which are likely to be effective in clinical trials with minimal side effects. These findings could significantly streamline drug development for SNHL, potentially reducing the costs and time associated with pharmaceutical research and development.

## 1. Introduction

Sensorineural hearing loss (SNHL) is a prevalent condition characterized by impaired sound perception and nerve impulse conduction in the auditory system. It affects a substantial number of individuals worldwide, with estimates from the World Health Organization (WHO) suggesting that around 466 million people currently suffer from disabling hearing loss [1,2]. Alarmingly, this number is projected to exceed 900 million by the year 2050. SNHL is the most common form of hearing loss and is a leading cause of disability globally [3], accounting for approximately 90% of reported cases [4]. Unfortunately, there is currently no specific clinical drug or therapy available to restore hearing in individuals with SNHL, leading conventional medicine to consider it irreversible. Consequently, SNHL research remains a challenging and vital topic within the academic community, as the search for understanding its pathogenesis and identifying therapeutic targets for differential proteins is of great significance.

In recent years, approaches to treating SNHL have evolved significantly. Previously, limited treatment options focused on symptom management, often relying on hearing aids to amplify sound and improve communication. However, modern treatments have become more diverse and refined, encompassing techniques such as intra-drum steroid therapy [5], hyperbaric oxygen therapy (HBOT) [6], and emerging regenerative therapies like hair cell regeneration, gene therapy, and the use of specific molecular needles [3]. Despite these advancements, challenges persist. Not all patients respond equally to the current treatments, and their effectiveness may vary based on factors such as the severity and onset of hearing loss. Moreover, regenerative therapies are still in the developmental stage and are not yet widely accessible. Complete restoration of hearing remains elusive even with the best available treatments.

To address these challenges and accelerate the discovery and validation of drug targets, we employ Mendelian randomization (MR) analysis. This approach utilizes human genetic data to identify causal relationships between predicted genetic traits, offering significant advantages over observational studies [7]. MR overcomes biases caused by confounding and reverse causation by providing a quasi-experimental approach akin to randomized trials. By leveraging genetic variants as proxies for interventions on drug targets, MR investigates their potential impacts on biomarkers and disease outcomes, thereby facilitating rapid application across various targets [8]. In this study, we recognize the importance of identifying effective therapeutic targets and their potential adverse effects. This not only aids in understanding the pathophysiological mechanisms of SNHL but also guides the development of new drugs, enhances the success rate of clinical trials, and reduces the time and cost of drug development. To achieve this goal, we employed MR analysis, a method that utilizes human genetic data to identify causal relationships. This approach can overcome the confounding factors and reverse-causality issues faced by traditional observational studies. Through this method, we can more accurately assess the safety and efficacy of candidate targets, thereby providing a solid foundation for future clinical applications [9].

This study was conducted by bioinformatics methods, and all the analyses were based on computer-simulated data and did not involve any clinical trials. We utilized MR analysis to identify potential therapeutic targets for SNHL. Initially, we employed MR and genome-wide association study (GWAS) data, along with plasma proteins from plasma protein quantitative trait loci (pQTL) data, to identify plasma proteins that may contribute to SNHL. Subsequently, we conducted a colocalization analysis to validate these preliminary findings. A phenotype-wide association analysis explored the association between potential therapeutic targets and other features, offering valuable insights into their versatility and potential mechanisms of impact. Additionally, we performed drug prediction and molecular docking studies to validate the pharmacological activity of SNHL drug targets, assess their feasibility, and identify potential drug candidates. Lastly, gene enrichment analysis deepens our understanding of SNHL development and mechanisms. Through this comprehensive approach, our study aims to contribute to the advancement of SNHL research and the identification of potential therapeutic strategies for this debilitating condition.

## 2. Materials and Methods

The pooled data of GWAS and pQTLs used in this study were obtained from published studies, and all the participants provided informed consent in their respective original studies. Therefore, no additional ethical approval was required. The research design is illustrated in Figure 1, and the specific research details are described below.

### 2.1. Plasma Protein Quantitative Trait Loci (pQTL)

The data for the pQTLs of plasma proteins were obtained from the UK Biobank Pharma Proteomics Project (UKB-PPP database) (https://www.synapse.org/#!Synapse:syn51365301) and accessed on 21 April 2024. This database encompasses data on the characterization of 2923 plasma protein pQTLs in 54,219 participants. It reveals the identification of 14,287 significant genetic association loci, with 81% of these loci being novel and not previously documented [10]. To enhance the selection of more effective instrumental variables (IVs), the following criteria were established: (1) genome-wide significance threshold of *p* < 5 × 10^−8^; (2) removal of IVs with an F-statistic less than 10 to mitigate bias arising from weak IV factors (F = (beta/se)2); (3) selection of loci for cis-pQTLs 1 Mb upstream and downstream of protein genes (cis-pQTL are mainly pQTLs close to the regulated protein genes and are generally mostly located within the 1 Mb region upstream and downstream of the regulated genes); and (4) reduction of linkage disequilibrium aggregation to ensure independent associations (r^2 < 0.001) [11,12].

### 2.2. SNHL GWAS Dataset

The data for a genome-wide association study (GWAS) on SNHL were sourced from the R9 version of the FinnGen database (https://r9.finngen.fi, accessed on 21 April 2024) [13]. The disease diagnosis and inclusion criteria were based on ICD-10 codes H90.3, H90.4, and H90.5, and ICD-9 code 3891. A total of 32,487 cases and 331,736 controls were included in the analysis for comprehensive data insights.

### 2.3. Mendelian Randomization Analysis

To assess for causal effects, we performed two-sample MR analyses between plasma proteins and SNHL using the R package “twosampleMR”, version 0.4.22 [14]. The R package’s built-in capabilities were used to import screened SNPs and SNHL data. For MR analysis, the SNPs followed three MR hypotheses: (1) the SNPs and plasma protein expression are closely related (*p* < 5 × 10^−8^); (2) there is no direct association between the SNPs and SNHL; and (3) the SNPs can only affect SNHL through plasma proteins [15].

To evaluate the causal relationship between plasma proteins and SNHL, we employed the Wald ratio as the causality assessment criterion for a single IV and the inverse variance weighting (IVW) method for scenarios involving more than two IV_S_ [16]. To assess the potential for instrumental-variable bias, the F-statistic was calculated. An F-value exceeding 10 indicates the absence of such bias. The F-statistic is computed using the formula F = beta^2^/se^2^ [17] (Appendix A). Due to the repeated nature of the calculations, we applied a false discovery rate (FDR) correction, considering Pfdr < 0.05 as statistically significant (Appendix A).

### 2.4. Colocalization Analysis

To ascertain whether plasma proteins and SNHL are influenced by the same genetic variants, we conducted a gene colocalization analysis. For plasma proteins exhibiting positive MR results, we specifically chose cis-pQTLs within a 1 MB region upstream and downstream of their respective genes. These cis-pQTLs were then subjected to gene colocalization analysis with GWAS data related to SNHL. Colocalization provides the posterior probability (PP) for five hypotheses on whether a single variant is shared between two traits. PPH0: SNPs in the selected position are not associated with either phenotype 1 or phenotype 2 disease; PPH1: SNPs in the selected position are associated with phenotype 1 and are not associated with phenotype 2 disease; PPH2: SNPs in the selected position are associated with phenotype 2 and are not associated with phenotype 1; PPH3: SNPs in the selected position are associated with both phenotype 1 and 2 but are two independent SNPs; PPH4: SNPs in the selected position are associated with both phenotype 1 and 2 and are shared SNPs [18]. In this study, due to the limited capacity of colocalization analysis, we focused on plasma proteins with an association posterior probability (PPH3 + PPH4) equal to or greater than 0.8 [19].

### 2.5. Phenotype-Wide Association Analysis

Phenome-wide association studies (PheWAS), often referred to as reverse GWAS, represent a methodological framework for examining the relationships between SNPs or phenotypes and a broad spectrum of phenotypes that encompass the entire phenome. This methodology proves particularly instrumental in assessing the potential adverse effects associated with drug targets. In the current investigation, the exposure was based on plasma proteins that demonstrated positive MR outcomes, while the criteria for selecting instrumental variables adhered to previously established standards. The analysis focused on acquiring phenotypic information from the Finnish database, specifically version R9, which includes a total of 2272 distinct phenotypes organized into 46 categories [13]. This comprehensive dataset was utilized to conduct phenome-wide MR analyses. A *p*_fdr_-value below 0.05 was considered statistically significant.

### 2.6. Functional Enrichment Analysis

To investigate the interactions between SNHL-related proteins and known disease targets, we searched for “sensorineural hearing loss” using the GeneCards database (https://www.genecards.org/, accessed on 21 April 2024). The GeneCards database was chosen due to its comprehensive integration of gene information from multiple authoritative sources, including the literature, experimental data, and other bioinformatics tools. The GeneCards scoring system quantifies the relevance of genes to diseases, providing a reliable basis for screening genes associated with SNHL. Additionally, GeneCards is frequently updated, ensuring the timeliness and accuracy of the data [20]. The GeneCards score is a composite metric that reflects the relevance of a gene to a specific disease or phenotype. A higher score indicates a greater amount of evidence supporting the association between the gene and the condition of interest. For our study, we selected protein-coding genes associated with sensorineural hearing loss (SNHL) and set a threshold of a score greater than 50 as the selection criterion. The scoring range is from 0 to 100, and a score of 50 represents a relatively stringent threshold. This ensures that the selected genes have substantial evidence linking them to SNHL. Next, Metascape was selected for the functional enrichment analysis because it simultaneously performs Gene Ontology (GO) and Kyoto Encyclopedia of Genes and Genomes (KEGG) pathway analyses, offering a comprehensive annotation platform. Compared to similar tools, Metascape stands out for its user-friendly interface and broad support for gene sets, making the analysis results more intuitive and easier to interpret [21].

### 2.7. Prediction of Candidate Drugs

To assess the suitability of target genes as potential therapeutic targets, it is crucial to explore protein–drug interactions. In this investigation, we will leverage the Drug Signatures Database (http://dsigdb.tanlab.org/DSigDBv1.0/, accessed on 21 April 2024). DsigDB serves as a comprehensive database linking drugs and other chemicals to their associated target genes. With 22,527 gene sets and 17,389 unique compounds spanning 19,531 genes, DsigDB provides an extensive resource for understanding the relationships between drugs and their molecular targets. This step is instrumental in evaluating the therapeutic potential of the target genes, shedding light on their suitability for clinical intervention [22]. A composite score was calculated by integrating the *p*-value, OR, and other bioinformatics indicators to evaluate the overall potential of each drug, and preference was given to drugs with lower *p*-values, indicating a more significant association with the target genes.

### 2.8. Molecular Docking

Molecular docking is a process wherein two or more molecules recognize each other and establish a stable composite structure through geometric and energy matching. In our study, drugs were subjected to molecular docking with screened plasma proteins, and the binding activity between the drugs and the plasma proteins was assessed based on the docking fraction value, also referred to as the affinity [23]. The SDF-format files of the drugs were obtained from the PubChem database (https://pubchem.ncbi.nlm.nih.gov/, accessed on 21 April 2024).) [24], while the PDB-format structures of plasma proteins were downloaded from the RCSB database (https://www.rcsb.org/, accessed on 21 April 2024).) [25]. The molecular docking operations were conducted using AutodockVina 1.2.2. Notably, an affinity value of <−4.25 kcal/mol suggests binding activity between the components, with values below <−5.0 kcal/mol indicating enhanced binding activity and those below <−7.0 kcal/mol pointing to robust docking activity between the two entities [26].

## 3. Results

### 3.1. We Found1908 Plasma Proteins Associated with SNHL in the MR Analysis

A total of 5458 cis-pQTLs and the corresponding 1908 plasma proteins were obtained by the above screening criteria for IVs. Combined with MR analysis, LATS1, TEF, LMNB2, OGFR, and EIF2AK3 were causally associated with SNHL. Specifically, elevated LATS1 (OR = 1.81; 95% CI, 1.39–2.38; *p* = 0.006) and LMNB2 (OR = 2.33; 95% CI, 1.55–3.51; *p* = 0.011) increased the risk of SNHL, whereas increased TEF (OR = 0.60; 95% CI, 0.47–0.78; *p* = 0.006), OGFR (OR = 0.65; 95% CI, 0.53–0.82; *p* = 0.027), and EIF2AK3 (OR = 0.77; 95% CI, 0.67–0.89; *p* = 0.048) decreased the risk of SNHL (Table 1 and Figure 2A).

### 3.2. Sensitivity Analysis for Five Plasma Proteins Associated with SNHL

The data suggest that proteins that yield favorable results from MR and colocalization analyses are more likely to be authorized as therapeutic targets [27]. Four plasma proteins—LATS1, TEF, LMNB2, and OGFR—with PPH3 + PPH4 > 0.8 were identified by colocalization analysis of the five plasma proteins that were screened by MR (Figure 2B and Appendix A).

### 3.3. Phenotype-Wide MR Analysis of Four Plasma Proteins in SNHL

The current study used the R9 version of the Finnish database, which contains 2272 phenotypes divided into 46 classes for phenotype-wide analysis, to further evaluate whether the four possible drug target proteins found would have positive or negative effects on other phenotypes. The whole-phenotype analysis results can be understood as the correlation between certain diseases or features and the expression of genetically determined proteins. This study’s results are further supported by the fact that none of the four medication targets showed any significant protein-level correlation with any of the other traits. This suggests that there is a small likelihood of any adverse effects from pharmaceuticals acting on these targets (Figure 2C and Appendix A).

### 3.4. Enrichment Analysis of GO Annotation and KEGG Pathway

GO enrichment analysis and KEGG enrichment analysis are important tools in bioinformatics, used to interpret gene expression data to reveal biological functions. GO enrichment analysis focuses on identifying the common characteristics of genes in gene sets in terms of biological processes (BPs), molecular functions (MFs), and cellular components (CCs). On the other hand, KEGG enrichment analysis concentrates on analyzing the role of gene sets in metabolic pathways and signal transduction [28]. We screened 54 genes related to SNHL from the GeneCards database (Appendix A) and conducted enrichment analysis together with plasma-positive proteins. As shown in Figure 3A, the main BPs that were enriched were hippo signaling, regulation of canonical Wnt signaling pathway, and vasculogenesis. In terms of CCs, the genes were mainly associated with cell–cell adhesion (cell–cell junction, microtubule-organizing center attachment site, focal adhesion, and cell–cell contact zone). Hair cells are connected to the surrounding cells and matrix by specialized cell adhesion molecules. This adhesion is necessary to maintain the structural stability and position of the hair cell and also influences the response of the hair cell to sound waves [29]. In terms of MFs, proteins were mainly focused on binding regulation (lamin binding, transcription factor binding, transmembrane transporter binding, I-SMAD binding, kinase binding, actin binding, and DNA-binding transcription activator activity, RNA polymerase II-specific). As shown in Figure 3B, KEGG enrichment results mainly in the regulation of cell proliferation and apoptosis (hippo signaling pathway, tight junction, and apoptosis).

### 3.5. Screening of Potential Therapeutic Drugs

In this study, the DsigDB database was used to screen drugs for the treatment of SNHL, and the top 10 drugs were selected based on the *p*-value (Table 2) [30]. The results showed that atrazine CTD 00,005,450 corresponded to the most therapeutic targets, followed by ampyrone HL60 DOWN, genistein CTD 00007324, and arsenenous acid CTD 00000922. In addition, LMNB2, LATS1, and OGFR could be interfered with by a variety of drugs; however, there were fewer TEF-intervening drugs.

### 3.6. Validation of Candidate Plasma Proteins in Treating SNHL

To further understand the affinity between drugs and target proteins, thereby gaining insights into the drugability of the target proteins, we conducted molecular docking studies. Due to the absence of the three-dimensional structure of OGFR in the PDB, we ultimately obtained molecular docking data for only six groups (Table 3 and Figure 4). The results indicate that there are predicted binding sites between the drug and the target protein, with an affinity of less than −7 kcal/mol, suggesting a strong interaction. Additionally, we performed molecular docking of the target proteins with dexamethasone, an existing treatment for SNHL, which showed an affinity ranging from −4 to −7 kcal/mol (Appendix A). This evidence supports the potential of these target proteins as therapeutic targets for SNHL.

## 4. Discussion

The development of novel therapeutic agents for SNHL is extremely challenging. A major reason for this dilemma is the unknown pathophysiology of SNHL. In this study, based on the cis-pQTL for blood protein druggability, we identified four druggable protein expressions that may affect SNHL outcome, demonstrating that four genes (LATS1, TEF, LMNB2, and OGFR) are causally associated with SNHL. Simultaneous co-localization provided further strong evidence. To further illustrate the potential side effects of the target proteins’ possible drugs, phenome-wide association analysis was also used. In addition, enrichment analysis was performed in this study to understand the biological significance of these drug targets. Finally, predictions and molecular docking of the drugs corresponding to these targets were performed in this study to further demonstrate the drug-forming value of these target genes.

Through gene enrichment analysis, we found that proteins associated with SNHL are primarily involved in biological processes such as the Hippo signaling pathway and cell–cell adhesion. The Hippo signaling pathway is a conserved signaling network that plays a critical role in regulating organ size, cell proliferation, apoptosis, and stem cell self-renewal. During inner ear development, this pathway is essential for maintaining the normal function of hair cells and supporting cells. Specifically, LATS1, one of the core kinases in the Hippo signaling pathway, is responsible for phosphorylating and inhibiting the YAP/TAZ transcription coactivators, preventing their nuclear entry and, thereby, modulating downstream gene expression to influence the balance between cell proliferation and apoptosis. When LATS1 function is compromised, it can lead to the aberrant activation of YAP/TAZ, resulting in cell cycle disruption and increased apoptosis, which may ultimately contribute to hearing impairment [31,32]. Cell–cell adhesion is a crucial mechanism for maintaining tissue structural stability and functional integrity. In the inner ear, hair cells connect to the surrounding support cells and matrix through specialized adhesion molecules, ensuring proper spatial positioning and responsiveness to external sound wave stimuli, while also influencing their survival status. For instance, types of intercellular junctions such as tight junctions and adherens junctions prevent ions and other substances from non-specifically crossing the epithelial barrier, thus ensuring the homeostasis of the inner ear microenvironment [33,34]. If cell–cell adhesion is impaired, it could lead to the disordered arrangement of hair cells, dysfunction of support cells, or even cell death, all of which are potential factors contributing to the occurrence of SNHL.

LATS1 (large tumor suppressor 1) plays a critical role in inner ear development and auditory function, primarily through its regulation of the Hippo signaling pathway. LATS1 phosphorylates YAP (Yes-associated protein), inhibiting its nuclear translocation and transcriptional activity, thereby regulating the proliferation, differentiation, and polarity of cochlear sensory cells and supporting cells, ensuring the normal development and functioning of inner ear cells [32]. Mechanistically, the loss of LATS1 leads to YAP dephosphorylation and its accumulation in the nucleus, activating downstream pro-proliferative and anti-apoptotic genes and disrupting the balance of the cell cycle and apoptosis. This results in a reduced numbers of sensory cells, disorganized cell arrangements, and increased apoptosis, ultimately leading to congenital hearing loss [35]. Additionally, LATS1 regulates auditory-related genes (e.g., genes involved in hair cell polarity) and signaling pathways (e.g., MAPK and NF-κB pathways), maintaining the normal transmission and processing of auditory signals [36]. Therefore, LATS1, by precisely modulating YAP activity and downstream signaling networks, is indispensable for inner ear development and auditory function. Its dysfunction may contribute to hearing loss and related pathological changes.

LMNB2 can regulate genes associated with cell proliferation, such as p21, cyclin D1, and cyclin E1, through direct or indirect mechanisms. These genes play crucial roles in cell cycle regulation and cell proliferation. For instance, LMNB2 promotes cell cycle progression by inhibiting the expression of p21, thereby enhancing cell proliferation capacity [37,38]. Simultaneously, LMNB2 facilitates the transition from the G1 phase to the S phase by modulating the expression of cyclins such as cyclin D1 and cyclin E1, thereby accelerating cell proliferation [39]. Additionally, LMNB2 is involved in regulating chromatin structure and stability, which is essential for gene expression regulation and the maintenance of cell proliferation. Recent studies have shown that the expression of LMNB2 in cochlear cells is closely related to the development of SNHL. The abnormal expression of LMNB2 may lead to cochlear cell dysfunction, thereby affecting hearing. Specifically, LMNB2 influences the survival and proliferation of cochlear cells by regulating cell cycle and apoptosis pathways [40]. In some studies, it has been found that the expression level of LMNB2 is positively correlated with the proliferative capacity of cochlear cells, and the absence of LMNB2 leads to increased apoptosis of cochlear cells, exacerbating the degree of hearing loss [37]. Furthermore, LMNB2 may also participate in the response of cochlear cells to external stimuli by affecting nuclear structure and function. Therefore, research on LMNB2 not only provides new insights into the mechanisms of sensorineural hearing loss but also offers potential targets for future therapeutic strategies.

OGFr is a receptor that binds to the opioid growth factor (OGF, [Met^5^]-enkephalin) and is widely distributed in the nervous system, where it participates in cell proliferation, tissue repair, and neural regulation [41]. The auditory system relies on complex neural signaling and cell proliferation processes, particularly the development and functional maintenance of inner ear hair cells and auditory neurons. OGFr may indirectly influence the development and function of the auditory system by regulating the proliferation and differentiation of neurons. Additionally, OGFr plays a significant role in lipid metabolism and energy balance, particularly in regulating lipid oxidation and thermogenesis in adipose tissue [42]. Inner ear hair cells and auditory neurons are highly sensitive to metabolic homeostasis, and metabolic disturbances may lead to hearing loss. OGFr may protect hearing by regulating lipid oxidation and energy metabolism, thereby maintaining the normal function of inner ear cells. OGFr also plays a role in immune regulation and inflammatory responses. Hearing loss may be associated with inflammation or immune responses in the inner ear, such as sudden sensorineural hearing loss or noise-induced hearing loss. OGFr may indirectly protect the auditory system from damage by modulating immune cell function or suppressing inflammatory responses [43,44]. For instance, the OGF-OGFr pathway has been found to inhibit inflammation and modulate immune responses in cancer therapy [45], providing a reference for its potential application in the auditory system.

TEF is a member of the PAR bZip gene family [46], which is involved in neurotransmitter homeostasis, amino acid metabolism, and apoptosis regulation [47,48]. In this study, apoptosis plays an important role in SNHL. The relationship between apoptosis and SNHL has been the subject of various studies, demonstrating the role of programmed cell death in hearing impairment. These studies explore different mechanisms of cell death, such as apoptosis, autophagy, and programmed necrosis, and their signaling pathways in the context of hearing loss. For instance, research on FGF13-knockout mice reveals that the mitochondrial apoptosis in cochlear spiral ganglion neurons leads to sensorineural hearing loss. Another study investigates the increased apoptosis in spiral ganglion neurons in apolipoprotein E-knockout mice under the influence of a Western diet, highlighting a link between dietary factors, atherosclerosis, and hearing loss. These findings suggest that genetic and environmental factors can influence the development of hearing loss [49,50,51]. Apoptosis can act as a mechanism of action for TEF and thus intervene in the treatment of SNHL.

Molecular docking is a computational technique used to predict the interaction between small molecules, such as drug candidates, and biomolecules like proteins. This technology aids in evaluating the potential binding capabilities of prospective drugs to target proteins, thereby inferring their possible efficacy. In our study, we not only measured the binding affinities of candidate compounds to the target protein but also referred to the binding profiles of several known effective drugs against the same target [52]. The results indicate that the binding affinity values of our candidates are comparable to those of these effective drugs, typically ranging from −7 to −10 kcal/mol, which is common among most successfully marketed drugs. This suggests potential therapeutic value for our candidates [53]. Moreover, we conducted an in-depth analysis of the binding modes of successful drugs and compared our candidate compounds against this benchmark. Such comparisons enhance our understanding of the candidates’ potential and provide concrete directions for subsequent research and development. Overall, these findings support the possibility of the further development of the candidate drugs. While molecular docking is a powerful predictive tool, it relies on static protein structure models, which may not account for dynamic changes under physiological conditions. Additionally, identifying the correct binding pocket can be challenging, potentially leading to results that do not fully reflect real biological interactions. We recognize these limitations and have approached data interpretation with caution to ensure the reliability of our conclusions.

A443654 is a small-molecule inhibitor primarily targeting the PI3K/Akt signaling pathway, which plays a critical role in the occurrence and development of various cancers, especially in cell proliferation, survival, and metabolic regulation. Research indicates that A443654 significantly reduces the phosphorylation levels of Akt by inhibiting the activity of PI3K, thereby blocking downstream signal transmission, leading to cell cycle arrest and apoptosis [54]. Genistein exerts its antitumor effects mainly through modulating multiple signaling pathways. Firstly, genistein can inhibit tumor cell proliferation and induce apoptosis by affecting the PI3K/Akt and MAPK/ERK signaling pathways. Studies have shown that genistein downregulates proteins associated with cell cycle regulation, thus suppressing the growth of tumor cells [55]. Nintedanib is an orally administered multi-targeted tyrosine kinase inhibitor primarily targeting vascular endothelial growth factor receptors (VEGFRs), platelet-derived growth factor receptors (PDGFRs), and fibroblast growth factor receptors (FGFRs) among several signaling pathways. The inhibition of these targets effectively halts the angiogenesis processes related to tumors, thereby suppressing tumor growth and metastasis. By interfering with the activation of these receptors, nintedanib decreases the proliferative and migratory capacity of tumor cells and improves the vascular structure within the tumor microenvironment, thus inhibiting tumor progression [56]. These findings suggest that the selected drugs may serve as potential agents for SHNL treatment.

This research has several key strengths. First, this is the first study to use the MR analysis technique to discover SNHL drug targets, using the most recent UKB-PPP plasma protein database available for drug target screening. The likelihood of false positives is considerably reduced, and so clinical trial success rates can be improved further. Enrichment study revealed the functional features of these genes. The final drug predictions demonstrate these genes’ medicinal potential, and the high binding activity of molecular docking implies that these genes have a high potential as drug targets. This study includes a comprehensive evaluation from identification to drug binding qualities, and it proposes four therapeutic candidates for SNHL with solid evidence.

While our research is based on large-scale population data, some proteins may have insufficient sample sizes to detect smaller effect sizes. Additionally, since the data originate from aggregated statistics of different populations, there might be population-specific effects or changes over time that cross-sectional studies cannot capture. Lastly, although we endeavored to select the strongest instrumental variables, we cannot entirely rule out horizontal pleiotropy, where an SNP may influence multiple unrelated biological processes [7,57].

Moreover, while our study has identified potential drug targets and candidate drugs through various bioinformatics methods, to effectively translate these findings into clinical applications, we recognize the need for systematic preclinical testing and validation. For this purpose, we plan to undertake the following steps: First, evaluate the effects of candidate drugs on the expression and function of LATS1, TEF, LMNB2, and OGFR using in vitro cell models. Second, test the safety and efficacy of the drugs in vivo using animal models, delving deeper into the relevant signaling pathways and mechanisms. Third, conduct comprehensive toxicology and pharmacokinetic studies to ensure drug safety. Finally, we will foster multidisciplinary collaboration to optimize the transition from preclinical to clinical trials, ensuring that research outcomes can smoothly progress to the clinical stage. These steps will help accurately assess the efficacy and safety of the identified drug targets and candidate drugs, laying a solid foundation for future clinical trials.

Despite the success of this study in identifying potential therapeutic targets and candidate drugs, to enhance the specificity of clinical application, we propose the following steps: Firstly, employ genetic testing technologies to conduct individualized screening for key genes such as LATS1, TEF, LMNB2, and OGFR and establish a genotype–phenotype correlation database for patients. Secondly, based on the patient’s genotypic information and results from drug sensitivity tests, develop personalized treatment protocols. Finally, during the course of treatment, regularly monitor the expression levels of target proteins to evaluate efficacy and adjust the treatment plan as necessary.

While this study successfully identified promising therapeutic targets for SNHL through various methods, we are aware of potential biases that may arise from the datasets and methodologies used. To mitigate these biases, we have taken the following measures: First, since the UKB-PPP and FinnGen databases primarily contain data from individuals of European ancestry, there is a potential issue of insufficient representation of other ethnic groups. Therefore, we emphasize in our result interpretation that the findings are applicable to populations with similar genetic backgrounds and call for future studies to include more diverse samples. Second, to minimize selection bias, we adopted stringent SNP screening criteria (e.g., *p*-value < 5 × 10^^−8,^ F-statistic > 10) to ensure the validity and independence of the selected genetic variants. These measures help enhance the reliability and broad applicability of our research findings.

## 5. Conclusions

This study aimed to explore potential drug targets in SNHL through the use of MR analysis. Four significant drug targets were identified in this study, and these were supported by co-localization analysis. The phenotype-wide analysis showed that the drugs acting on the above drug targets had fewer side effects. In addition, this study further validated the drug potential of these targets through drug prediction and molecular docking techniques. These findings provide promising clues for the more effective treatment of SNHL, which may help reduce drug development costs and advance personalized treatment strategies.

## Figures and Tables

**Figure 1 bioengineering-12-00126-f001:**
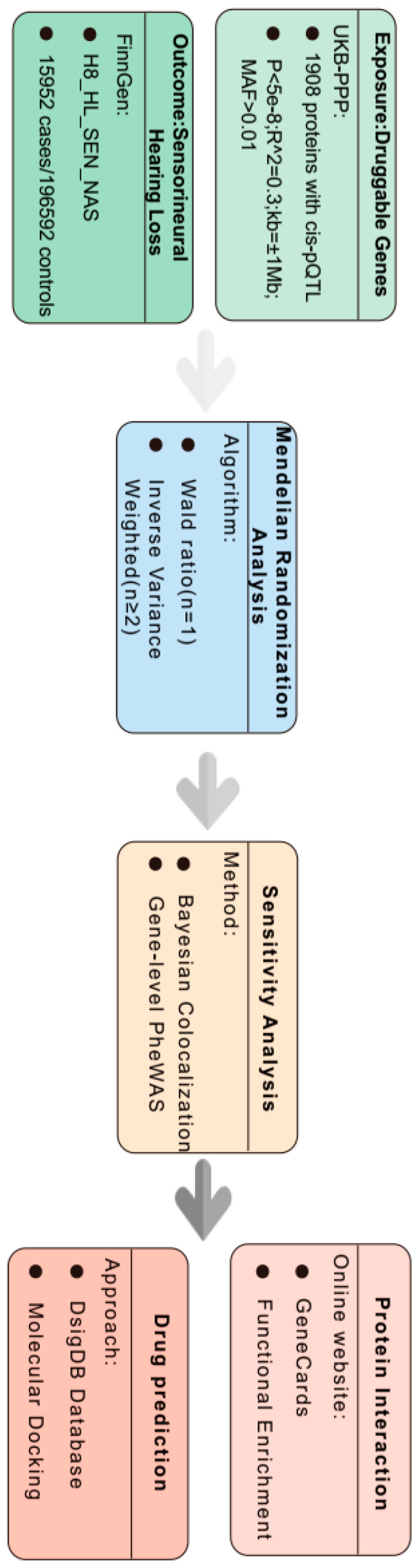
A study design for the identification of plasma proteins that are causally associated with sensorineural hearing loss (SNHL).

**Figure 2 bioengineering-12-00126-f002:**
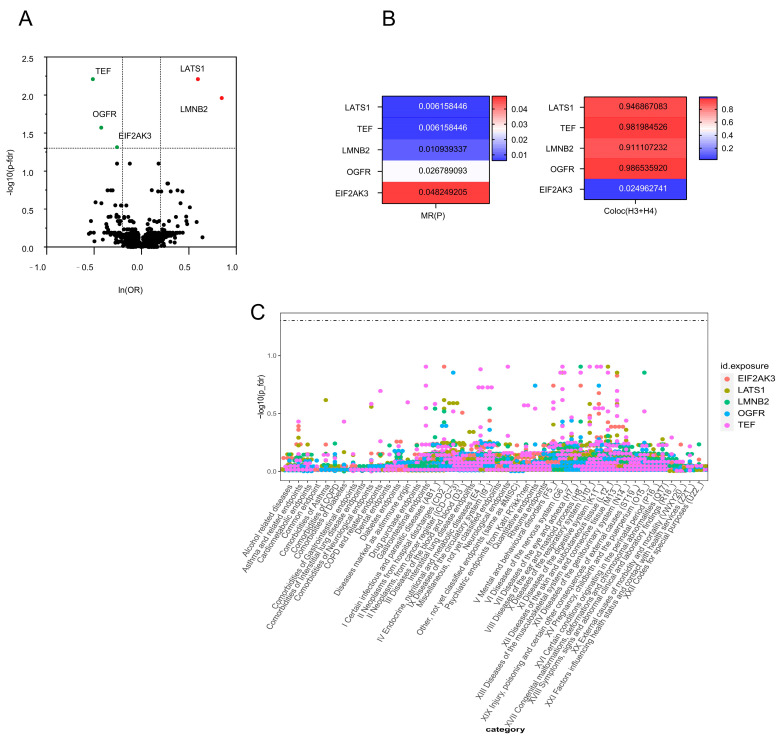
Drug target protein screening and phenotype-wide analysis. (**A**) MR analysis results for plasma proteins and risk of SNHL. The OR for increased SNHL risk was calculated based on the standard deviation (SD) increase in plasma protein levels. Statistically significant associations are represented by colored circles. (**B**) Colocalization analysis results for plasma proteins with significant MR findings. Each bar represents the posterior probability (PPH3 + PPH4) that the same genetic variant influences both the plasma protein and SNHL. (**C**) Manhattan plot illustrating phenotype-wide MR results for LATS1, TEF, LMNB2, and OGFR. Each dot represents a phenotype tested against the drug targets; significant associations are marked with different colors. Reference lines indicate the significance thresholds.

**Figure 3 bioengineering-12-00126-f003:**
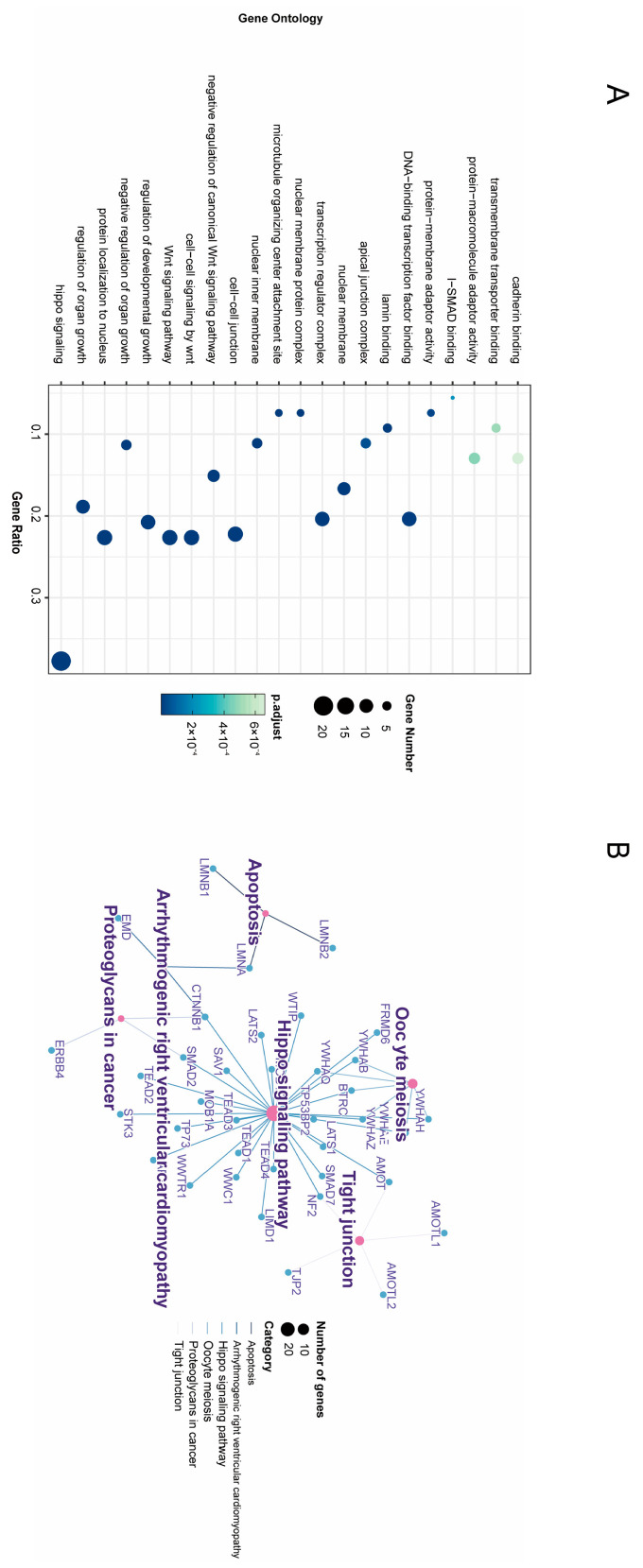
Functional enrichment analysis. (**A**) Results of Gene Ontology (GO) enrichment analysis across three categories. (**B**) Results of KEGG (Kyoto Encyclopedia of Genes and Genomes) enrichment analysis clustered and color-coded by cluster-ID. The factors on the X-axis are derived from standard lists in GO and KEGG databases to ensure a comprehensive and unbiased overview of potential biological mechanisms.

**Figure 4 bioengineering-12-00126-f004:**
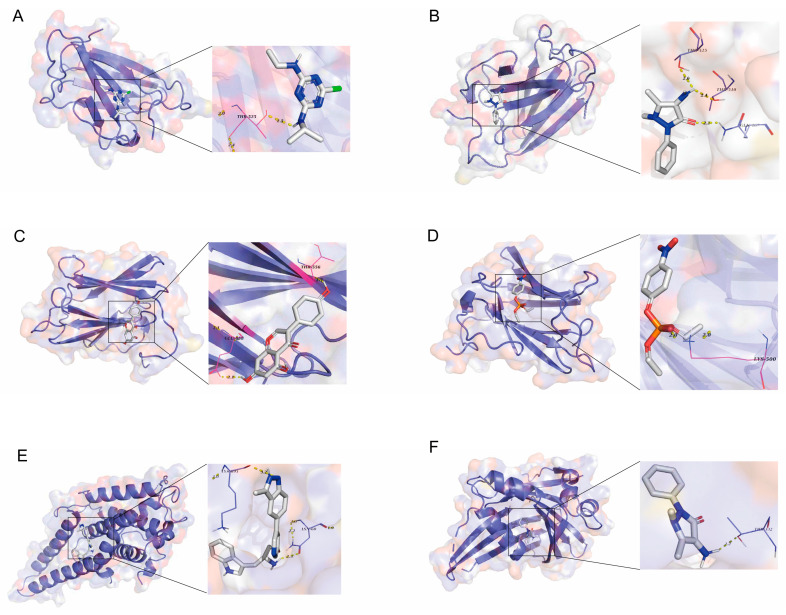
Molecular docking results for small-molecule drugs and target proteins. (**A**) LMNB2 docking with atrazine, (**B**) LMNB2 docking with ampyrone, (**C**) LMNB2 docking with genistein, (**D**) LMNB2 docking with PARAOXON, (**E**) LATS1 docking with A443654, (**F**) TEF docking with ampyrone.

**Table 1 bioengineering-12-00126-t001:** Forest plot of five plasma proteins significantly associated with SNHL in MR results.

Characteristics	Total (N)	HR (95% CI)	*p* Value
LATS1	1	1.81 (1.39–2.38)	0.006
TEF	1	0.60 (0.47–0.78)	0.006
LMNB2	1	2.33 (1.55–3.51)	0.011
OGFR	1	0.65 (0.53–0.82)	0.027
EIF2AK3	1	0.77 (0.67–0.89)	0.048

**Table 2 bioengineering-12-00126-t002:** Candidate drugs predicted by the DSigDB database.

Drug	*p*-Value	Odds Ratio	Combined Score	Genes
Ampyrone HL60 DOWN	0.009	24.669	117.1276	OGFR; LMNB2
Atrazine CTD 00005450	0.012	17.2329	76.7860	OGFR; TEF; LMNB2
PARAOXON CTD 00006470	0.014	93.5449	397.178	LMNB2
A443654 LINCS	0.021	64.379	249.824	LATS1
Genistein CTD 00007324	0.021	15.270	59.0690	OGFR; LMNB2
Arsenenous acid CTD 00000922	0.023	14.610	55.357	OGFR; LMNB2
SU-14813 Kinome Scan	0.029	45.634	161.7507	LATS1
AST-487 Kinome Scan	0.029	45.009	158.928	LATS1
Nintedanib FDA	0.031	42.121	146.008	LATS1
BIBF-1120 (derivative) Kinome Scan	0.031	42.121	146.0084	LATS1

**Table 3 bioengineering-12-00126-t003:** Molecular docking results for drug and target proteins.

Target	PDB ID	Drug	Binding Energy (kcal/mol)
LMNB2	2LLL	Atrazine	−63.766
LMNB2	2LLL	Ampyrone	−67.623
LMNB2	2LLL	Genistein	−86.926
LMNB2	2LLL	PARAOXON	−67.057
LATS1	7LWH	A443654	−7.949
TEF	8CAA	Ampyrone	−7.836

## Data Availability

The original contributions presented in the study are included in the article/Appendix A; further inquiries can be directed to the corresponding authors.

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
