# Peer review of "Identification of Potential Therapeutic Targets for Sensorineural Hearing Loss and Evaluation of Drug Development Potential Using Mendelian Randomization Analysis"

_bioengineering, 2025, doi:10.3390/bioengineering12020126_

Round 1

Reviewer 1 Report

Comments and Suggestions for Authors

Summary

Mendelian randomization analysis and genome-wide association studies are used in this paper to find possible treatment targets for sensorineural hearing loss. Four important proteins—LATS1, TEF, LMNB2, and OGFR—are identified as interesting pharmacological targets in the study. Their molecular mechanisms are investigated using enrichment analysis, and treatment candidates are assessed using molecular docking.

Strengths

The integration of MR analysis, GWAS, and molecular docking offers a novel and efficient framework for identifying therapeutic targets in SNHL.

The study covers multiple layers of target validation, from genetic association to functional enrichment and drug screening.

Identifying druggable targets with minimal side effects addresses a critical unmet need in SNHL treatment.

Recommendations

The study relies on MR and colocalization for target identification but provides limited validation of statistical robustness. Could the authors include metrics like the F-statistic for instrumental variable strength and sensitivity tests for pleiotropy?

Figures 2 was cluttered and lack clear annotations. Simplify the visuals and include legends with concise explanations of key results. The Manhattan plot (Figure 2C) is visually dense and hard to interpret. A supplementary table summarizing phenotype-wide associations would improve accessibility.

While LATS1 and LMNB2 are linked to apoptosis and cell proliferation, their specific roles in SNHL are not fully discussed. Could the authors expand on how these proteins mechanistically contribute to auditory function and damage?

Could the authors compare the docking affinities of the identified drugs with known SNHL treatments to strengthen their claims?

The study lacks experimental validation of the proposed targets and drugs. Incorporating in vitro or in vivo experiments to confirm the functional roles of LATS1, TEF, LMNB2, and OGFR would significantly enhance the manuscript’s impact.

While drug candidates are identified, the manuscript does not outline a clear roadmap for translating these findings into clinical practice. Could the authors provide recommendations for further preclinical testing and validation?

The study concludes with a broad suggestion for clinical relevance but lacks specificity. Could the authors propose concrete steps for integrating these findings into personalized medicine approaches for SNHL?

A thorough proofreading pass is recommended to address minor grammatical issues and improve overall readability.

Conclusion

With important clinical and research ramifications, the paper offers a strong computational paradigm for SNHL therapeutic target identification. However, a lack of experimental evidence, ambiguous representations, and inadequate validation restrict its impact. The publication will be a useful addition to the fields of bioengineering and hearing research if these problems are resolved.

Reviewer 2 Report

Comments and Suggestions for Authors

The paper titled "Identification of Potential Therapeutic Targets for Sensorineural Hearing Loss and Evaluation of Drug Development Potential Using Mendelian Randomization Analysis" presents a valuable exploration into the therapeutic avenues for sensorineural hearing loss (SNHL) using Mendelian Randomization (MR). Here are some major comments regarding the study:

1.        The study addresses an important gap in the treatment of SNHL, which is often overlooked in clinical practice. By identifying potential therapeutic targets, the research could pave the way for new treatment options, making it highly relevant to both clinical and pharmaceutical fields.

2.        The use of a robust database like UKB-PPP for cis-protein quantitative trait locus data and the FinnGen database for SNHL data enhances the credibility of the findings. The application of colocalization analysis and phenotype-wide association analysis is commendable, as it provides a comprehensive approach to understanding the relationship between genetic factors and drug targets.

3.        The identification of four significant drug target proteins (LATS1, TEF, LMNB2, and OGFR) is a notable achievement. However, it would be beneficial to provide more detailed information on how these proteins specifically relate to the pathophysiology of SNHL and their mechanisms of action.

4.        The study highlights that these targets have fewer potential side effects, more detailed information on how these conclusions were drawn would strengthen the findings. A deeper exploration into the side effect profiles of the identified drug targets would enhance the clinical applicability of this research.

5.        The methodology for screening drugs using DsigDB and molecular docking is appropriate, but additional details on the criteria for selecting the top drugs would be helpful. Including information on how these drugs were validated or any preliminary efficacy data could provide further support for their therapeutic potential.

6.        The use of chi-square analysis to assess relationships between drug reconciliation and drug incompatibility is appropriate; however, providing more context on the statistical methods used and their implications would enhance clarity. Additionally, discussing any limitations related to sample size or data sources could provide a more balanced view.

7.        The conclusion emphasizes that these findings could streamline drug development for SNHL, which is promising. However, discussing potential challenges in translating these findings into clinical practice or regulatory pathways would provide a more comprehensive overview.

8.       The methodology is outlined, it lacks sufficient detail in certain areas. For example, when discussing the use of the GeneCards database and the criteria for selecting proteins with scores >50, it would be beneficial to elaborate on what these scores represent and how they were determined. This would enhance reproducibility and provide readers with a clearer understanding of the selection process.

9.       The choice of databases such as GeneCards, Metascape, and DsigDB is appropriate; however, the rationale behind selecting these specific resources could be more explicitly stated. Discussing alternative databases or methods that were considered and why they were ultimately not chosen would provide a more comprehensive view of the research design.

10.   The explanation of molecular docking is somewhat technical and may benefit from simplification for clarity. Additionally, while binding affinity values are provided, there is no discussion on how these values compare to known standards in drug development or their implications for therapeutic efficacy. Including a comparison with existing drugs or known effective interactions could contextualize the findings better.

11.    The text does not address any limitations associated with the methods used, particularly regarding molecular docking and the assumptions made during protein-drug interaction predictions. Acknowledging potential biases or limitations in data interpretation would enhance the transparency and credibility of the research.

12.   Statistical significance is mentioned regarding binding affinities, there is no clear explanation of how statistical analyses were conducted to support these findings. Providing more details about the statistical methods used to evaluate results would improve the rigor of the analysis.

Reviewer 3 Report

Comments and Suggestions for Authors

1. The introduction should establish the importance of identifying both therapeutic targets and understanding their adverse effects.

2. Discuss any biases that might arise from the datasets or methods and how they were mitigated.

3. Expand on the implications of involvement in the Hippo signaling pathway and cell-cell adhesion.

4. Discuss the practical steps for moving from preclinical studies to trials.

5. Suggest ways to validate findings, e.g., through experimental or longitudinal studies.

Reviewer 4 Report

Comments and Suggestions for Authors

In this bioinformatic study, these workers have used Mendelian Randomisation (MR) to identify novel targets and repurpose drugs which could potentially treat sensorineural deafness hearing loss (SNHL). They mined the UK Biobank PPP database to access plasma cis-protein quantitative trait loci (cis-pQTL) data and used SNHL data from the FinnGen database as the endpoint for their MR causal analysis of drug targets. Co-localization analysis was used to detect SNPs that were common for influencing SNHL risk and protein expression while a phenotype-wide association analysis (PPA) was conducted to assess potential side effects of drugs targeting the identified protein receptors. Molecular docking was subsequently used to evaluate the binding energy of drugs predicted to be ligands at the identified targets. Four plasma protein drug targets were identified -  LATS1, TEF, LMNB2, and OGFR - and PPA predicted few potential side effects from ligands binding to these target proteins. The genes that were associated with SNHL were primarily active in the Hippo signaling pathway and were involved in cell-cell adhesion and the regulation of cell proliferation and apoptosis. Potential  repurposable drugs for SNHL treatment were found by screening the DsigDB database and using molecular docking software. Atrazine, ampyrone, genistein,  and paraoxon  and six others were identified.  It is concluded that this bioinformatic approach can successfully identify promising new drug targets for SNHL, and drugs are likely to be effective in clinical trials and have minimal side effects.

This is a novel study using state of the art bioinformatic approaches for idenifying novel targets and drugs which could potentially be repurposed to treat SNHL but I have some difficulties with the outcomes presented here:

First, molecular docking studies have a high failure rate when drugs predicted to bind to targets are tested in vivo as, although the drugs may bind well to the target, they often fail to be functionally active.

Second, these workers used a phenotype association analysis to find drugs that would be unlikely to have adverse side effects. The problem is that three of  the four drugs they finally  recommended for treating SNHL are highly toxic to humans: atrazine is a synthetic herbicide used on golf courses which can kill worms, bees, and aquatic life; ampyrone can cause agranulocytosis  and is no longer used as an analgesic; paraoxon is an organophosphate insecticide; genistein is a phytoestrogen and, though the least toxic, is used to kill intestinal worms and as an anti-cancer agent.

In summary, while the bioinformatic approach these workers have used to identify novel targets and treatments fo SNHL is of great interest, it is clear to me that bioinformatics alone cannot be used for drug discovery but needs to be backed up with in vitro and in vivo safety and efficacy studies.

Round 2

Reviewer 1 Report

Comments and Suggestions for Authors

By strengthening statistical validation, raising figure clarity, extending mechanistic explanations of target proteins, and comparing drug docking affinities with current treatments, the authors have fully addressed the  comments from previous round. They suggested actions for incorporating findings into personalized therapy and offered a clear road map for preclinical validation and clinical translation. They also fixed grammatical errors to make the text easier to read. The work has been much reinforced by these extensive edits, and it is now prepared for publishing.

Reviewer 2 Report

Comments and Suggestions for Authors

The authors have addressed each of my concerns thoroughly. They have provided clear explanations and made necessary revisions that enhance the clarity and quality of the manuscript. I appreciate their responsiveness to feedback, which has significantly improved the overall presentation of their work.

Reviewer 3 Report

Comments and Suggestions for Authors

I consider that the article may be published in its current form. 

Reviewer 4 Report

Comments and Suggestions for Authors

None